# Comparative Proteomics Study of Yak Milk from Standard and Naturally Extended Lactation Using iTRAQ Technique

**DOI:** 10.3390/ani12030391

**Published:** 2022-02-07

**Authors:** Mingxing Cao, Lin Huang, Suyu Jin, Mengbo Zhao, Yucai Zheng

**Affiliations:** College of Animal and Veterinary Science, Southwest Minzu University, Chengdu 610041, China; 190906012006@stu.swun.edu.cn (M.C.); huanglin200426@swun.edu.cn (L.H.); syjin65@163.com (S.J.); zmb0158@163.com (M.Z.)

**Keywords:** yak, milk, prolonged lactation, proteomics, iTRAQ

## Abstract

**Simple Summary:**

To elucidate the differences in milk protein compositions and mammary gland functions between yaks of standard lactation (TL yaks) and prolonged lactation (HL yaks), iTRAQ technique was used to compare the skim milk proteins in the two yak groups. A total of 202 differentially expressed proteins (DEPs) were revealed, among which 109 proteins were up-regulated and 93 were down-regulated in the milk of HL yaks compared to TL yaks. The bioinformatics analysis revealed that the differences in skim milk protein between the HL yaks and the TL yaks suggests that the mammary gland of the HL yak is at a degeneration stage.

**Abstract:**

Extended lactation is a common phenomenon in lactating yaks under grazing and natural reproduction conditions. To elucidate differences in milk protein compositions and mammary gland functions between yaks of standard lactation (TL yaks) and prolonged lactation (HL yaks), whole milk samples of TL yaks and HL yaks (*n* = 15 each) were collected from a yak pasture at the northwest highland of China. The iTRAQ technique was used to compare the skim milk proteins in the two yak groups. A total of 202 differentially expressed proteins (DEPs) were revealed, among which 109 proteins were up-regulated and 93 were down-regulated in the milk of HL yaks compared to TL yaks. Caseins including κ-casein, αs1-casein, αs2-casein, and β-casein were up-regulated in HL yak milk over 1.43-fold. The GO function annotation analysis showed that HL yaks produced milk with characteristics of milk at the degeneration stage, similar to that of dairy cows. KEGG enrichment showed that the metabolic pathways with the most differences are those that involve carbohydrate metabolism and the biosynthesis of amino acids. The present results highlight detailed differences in skim milk proteins produced by HL yaks and TL yaks and suggest that the mammary gland of HL yak is at the degeneration stage.

## 1. Introduction

The yak is one of the main unique livestock species in ethnic minority areas of the Qinghai–Tibet Plateau in China [1]. It provides important living materials for local farmers, with milk being one of the major products. There exist some differences in lactation between yaks and dairy cows. The lactation of dairy cows can be extended beyond the standard 305-d lactation through several manipulated strategies. Moreover, these cows are subject to a substantial decline in milk yield and their milk composition changes during the extended lactation stage [2]. Since the feeding and management mode in yaks is natural grazing and reproduction, they are completely dependent on natural grass resources and natural reproduction. Furthermore, considerable proportions of lactating yaks (about 15%) will experience a prolonged lactation stage if they are not pregnant during the calving year [3]. If these yaks are calved in the spring and milked for consumption, can maintain lactation in the following winter by calf suckling only, and can be milked again during the next summer season when there is enough grass available. Thus, they are at a stage of naturally prolonged or extended lactation, and yaks at this lactation stage are called half-lactating (HL) yaks—respective to the standard lactation of yaks, which are called total-lactating (TL) yaks. HL yaks exhibit their second lactation peak during the summer season. Due to the relatively large number of HL yaks, their milk constitutes an important part of yak milk resources.

The composition of milk from the extended lactation stage has been studied in dairy cows, and it was found that milk protein and fat concentrations increase significantly [4], which is probably related to the decline in milk yield. Compared to TL yaks, the milk yield of HL yaks decreases significantly, while the concentrations of milk protein and fat increase significantly [5]. Our previous study has discovered some differences in milk composition between TL yaks and HL yaks. For example, lactose level and α-lactalbumin percentage are reduced in HL yak milk [6], which reflects the composition characteristics of milk from late lactation [7]. However, the composition of HL milk has not been studied extensively, and the state of the mammary glands of HL yaks remains unclear.

Most of the bovine milk proteins, including caseins, α-lactalbumin, and β-lactoglobulin, are synthesized by the epithelial cells of the mammary gland, while a small part of milk proteins, such as immunoglobulins and albumin, are derived from the blood [8]. Therefore, the protein composition of milk is related to the function of the mammary gland to some extent—a well-known example is the high immunoglobulin level in bovine colostrum [9]. Isobaric tags for relative and absolute quantitation (iTRAQ)-based proteomics is a widely used tool to study the overall protein profile of cells or tissues [10].It is highly precise and has been used in the field of biomedicine, zoology, botany, and microbiology research. Reinhardt used the iTRAQ technique to analyze milk fat globules of membrane proteins and found that 26 proteins were up-regulated, while 19 proteins were down-regulated, in the mature milk compared to colostrum [11]. Zhang L. reported that during the first 9 days after calving, one-third of the proteins, especially the content of immunoglobulin, decreased significantly in the milk [12]. A review article by Roncada P. reported the last progress in proteomic analysis of milk from farm animals [13]. The objectives of the current experiments were to examine the detailed differences in milk proteins between TL and HL yaks and to elucidate the nutritional value of HL milk. Moreover, the current experiments investigated possible functional changes in the mammary glands of HL yaks, which is of significance for the utilization of yaks at this special lactation stage. 

## 2. Materials and Methods

### 2.1. Experimental Yaks and Milk Sampling

Milk samples of yaks were collected in August. The experimental yaks were raised on a pasture about 3500 m above sea level in Hongyuan county, Sichuan Province, China. The experimental yaks included TL yaks (*n* = 15) and HL yaks (*n* = 15). The TL yaks calved during the spring season (March to May), while the HL yaks calved one year earlier. Their milk secretion was maintained by calf suckling during the winter. All the experimental yaks were 4 to 7 years old and 2 to 4 parities. The experimental yaks and their calves were grazed on the same natural grassland during the daytime and were separated from their calves at night. The lactating yaks were milked by the hands of local farmers in the morning. Approximately 50 mL of whole milk were collected from each yak, which was transferred to the laboratory using dry ice and stored at −80 °C until analysis. All animal care and milking procedures were approved by The Animal Ethics Committee of Southwest Minzu University (No. swun20200138).

### 2.2. iTRAQ Analysis of Skim Milk Proteins of Yaks

The whole milk of each yak was centrifuged at 800× *g* and 4 °C for 20 min to prepare skim milk. The pooled skim milk samples of TL yaks and HL yaks were prepared by mixing equal volumes of 15 skim milk samples of corresponding yaks, respectively. The two pooled samples (2 mL each) were transported using dry ice to Shenzhen BGI Technology Co., Ltd. for iTRAQ analysis.

For the iTRAQ assay, the pooled skim milk samples were centrifuged at 25,000× *g* for 20 min to remove residual fat and cell debris. The supernatant skim milk was removed and mixed with 5 volumes of cold acetone and stored at −20 °C overnight. The mixture was centrifuged again, and the resulting pellet was used for further preparing a protein solution [14]. A total of 100 μg of protein from this solution was digested with Trypsin Gold (protein: trypsin = 20:1) at 37 °C for 12 h. The resulting peptides were labeled using the iTRAQ Reagent 8-plex Kit according to the manufacturer’s protocol, followed by fractionation using a Shimadzu LC-20AB HPLC equipped with a 4.6 mm × 250 mm Gemini C18 column (Phenomenex) [15]. The eluted peptides were pooled as 20 fractions and were then vacuum-dried, dissolved, and loaded on an LC-20AD nano HPLC (Shimadzu, Kyoto, Japan) equipped with a 2 cm C18 trap column. Then, the peptides were eluted into an 18 cm analytical C18 column. Mass spectrometry analysis was performed as described in previous studies [13]. Data was acquired using a TripleTOF 5600 System fitted with a Nanospray III source (AB SCIEX, Downtown Redwood City, America).

### 2.3. Bioinformatics Analysis

IQuant software was applied to the quantification of proteins. Proteins with a 1.2-fold change and a Q-value of less than 0.05 were determined as differentially expressed proteins, and they must be defined in at least 1 replicate experiment. All proteins with a false discovery rate (FDR) of less than 1% proceeded with the following analysis, including Gene Ontology (GO), Clusters of Orthologous Groups (COG), and Kyoto Encyclopedia of Genes and Genomes (KEGG) pathway. 

The KEGG database (http://www.genome.jp/kegg/, accessed on 6 November 2021) and the COG database (http://www.ncbi.nlm.nih.gov/COG/, accessed on 6 November 2021) were used to classify and group the identified proteins. Functional annotations of the proteins were performed using the Blast2GO program against the non-redundant protein database in NCBI (www.ncbi.nlm.nih.gov, accessed on 6 November 2021). The pathway analysis was carried out by KEGG (http://www.genome.jp/kegg/, accessed on 6 November 2021) [16]. 

## 3. Results

### 3.1. Identification of Skim Milk Proteins of Yaks

iTRAQ analysis identified 767 proteins in the skim milk samples of TL and HL yaks based on the 1930 unique peptides obtained (Appendix A). The protein coverage was between 0.001 to 0.999, of which 37.1% was identified as using at least two unique peptides. The length of most acquired peptides ranged between 7 and 17 amino acids (Figure 1A), and the molecular weights of approximately 80% of the identified proteins were less than 100 kDa (Figure 1B).

### 3.2. GO and COG Annotation of Identified Proteins

All identified proteins in the skim milk samples of TL and HL yaks were subjected to the GO analysis and were categorized into biological processes, cellular components, and molecular functions, based on their GO annotations (Figure 2, Appendix A). The proteins identified were classified by molecular functions and were enriched in 13 different functional terms. The majority were related to binding and catalytic activity (471 and 276 proteins in total, respectively), followed by structural, molecular activity, transporter activity, and enzyme regulator activity. The GO cellular location classifications of the proteins were involved in 16 categories, and the most obvious differences were in the cell, cell part, organelle, and organelle part. There were 23 biological process categories in which the identified proteins were involved, and the largest number of proteins were observed in the cellular process (475), followed by the metabolic process (388), the single-organism process (385), and biological regulation (299).

COG annotation of all the identified proteins revealed 24 functional categories (Figure 3, Appendix A). Among these COG categories, the skim milk samples of TL and HL yaks were highly enriched in several major functional COG categories, including post-translational modification, protein turnover, chaperones, energy production and conversion, cytoskeleton, carbohydrate transport, and metabolism. 

### 3.3. Differentially Expressed Milk Proteins of Yaks

A total of 202 differentially expressed proteins (DEPs) were revealed according to the standards of fold-change ratios ≥ 1.2 and *p* < 0.05, among which 109 proteins were up-regulated and 93 proteins were down-regulated in the milk of HL yaks compared to TL yaks (Appendix A). The top 50 DEPs with at least 2.18- and 1.54-fold change, respectively, are listed in Table 1 and Table 2. The 14-3-3 protein theta showed the largest fold change (increased approximately 10-fold in HL yak milk). Another 11 proteins were up-regulated by approximately 4-fold, including spliceosome RNA helicase BAT1, dolichyl-diphosphooligosaccharide-protein glycosyltransferase 48 kDa subunit, small nuclear ribonucleoprotein, protein S100-A1, tubulin beta-7, 40S ribosomal protein S3, glycogen phosphorylase, and hemoglobin subunit beta (Table 1). Four kinds of caseins were identified, including κ-casein, αs1-casein, αs2-casein, and β-casein. Compared with those of TL yaks, these caseins were up-regulated in HL yaks by 2.04-, 2.48-, 2.35-, and 1.43-fold, respectively. The enzyme γ-glutamyltransferase2, vitamin D-binding protein, retinol-binding protein 4, lactotransferrin, serotransferrin, keratin, cysteine-rich secretory protein 3 precursor, and vinculin were also up-regulated in the milk of HL yaks (Table 1, Appendix A).

### 3.4. GO Enrichment Analysis of DEPs

GO enrichment analysis of the 202 DEPs between TL and HL yaks demonstrated that 13, 19, and 22 protein categories were highly enriched in the cellular component, molecular function, and biological process categories with *p* < 0.05 or *p* < 0.01, respectively (Figure 4). Among these, the vesicle, cytoplasmic vesicle, and membrane-bounded vesicle were the most abundant categories in the cellular component (Figure 4A, Appendix A). The transporter activity, enzyme regulator activity, and enzyme inhibitor activity were the most abundant categories in the molecular function (Figure 4B, Appendix A). In the biological process, as many as 65 GO terms were enriched with *p* < 0.05, and the 22 major categories (*p* < 0.01) were related to oxidation-reduction processes and the generation of precursor metabolites and energy (Figure 4C, Appendix A).

### 3.5. Pathway Enrichment Analysis of DEPs

The 202 DEPs were used for pathway enrichment analysis, and 25 major pathways were highly enriched via KEGG with *p* < 0.05 (Appendix A). In addition, a scatter plot for the top 20 of KEGG enrichment results is shown in Figure 5. The KEGG pathways that the DEPs mainly participated in were: primary immunodeficiency, staphylococcus aureus infection, cytokine-cytokine receptor interaction, and dilated cardiomyopathy. The metabolic pathways with the most differences involved carbohydrate metabolism, the biosynthesis of amino acids, and fat digestion and absorption.

## 4. Discussion

Our previous research discovered some differences in milk between TL yaks and HL yaks [6]. For example, the contents of protein and fat and the activities of several enzymes, such as γ-glutamyltransferase and alkaline phosphatase, in HL yak milk were significantly higher compared to the milk of TL yaks. It has been reported that the yield and composition of milk are influenced by many factors, including human maternal age, time of delivery and maternal diet, and the stage of lactation, which was the most influential factor [17]. The detailed composition and quality of milk from extended lactation have been studied in cows and humans. One study found higher protein and fat concentrations, an unaffected casein to protein ratio, and protein composition of the bovine milk from the extended lactation [18]. Czosnykowska-Łukacka et al. reported that the concentration of carbohydrates in mother’s milk showed a negative correlation with lactation of about two years, while fat and protein concentrations were opposite. Moreover, during prolonged lactation in humans (over 18 months), it was found that the concentration of carbohydrates significantly decreases, and fat and protein concentration significantly increases [19]. The analysis of the composition of prolonged yak milk allows one to assess the nutritional value of milk. In this study, the milk protein profiles of TL yak and HL yak were intensively studied based on the iTRAQ technique. The expressions of some major milk proteins in HL yaks were increased, such as κ-casein, αS1-casein, αS2-casein, and β-casein. In addition, vitamin D-binding protein, retinol-binding protein 4, lactotransferrin, and serotransferrin levels in HL yak milk also increased (Appendix A). Proteins are the major nutrients of milk and have many functions except for providing proteins for nutrition purposes. The micronutrients in milk can also affect its function [20]. The present results indicate that HL yaks can provide more nutrients in milk compared to TL yaks.

Based on their GO functional annotations, all identified proteins in the skim milk samples of TL and HL yaks were classified in accordance with molecular function, cellular localization, and biological pathways. It was reported that 66% of the DEPs identified in whey from yak colostrum and mature milk were found to be related to binding activity [21]. Approximately 44% and 22.4% of the identified proteins in bovine colostrum were involved in catalytic activity and binding activity, respectively [22]. Our present results are basically consistent with these reports, and the catalytic activity and binding activity account for the largest proportion of the identified proteins (Figure 2).

The mammary gland undergoes morphological and functional changes during development. The lactation cycle of cows includes early lactation, middle lactation, late lactation, and dry lactation [23]. Mammary gland degeneration is a key stage in dry lactation, and during this stage, the ability to synthesize milk decreases and cell apoptosis increases [4]. It was shown that the concentration of lactoferrin in mammary secretions during the dry period was significantly increased, which could be used to measure the degree of mammary degeneration in dairy cows [24]. In this study, the expressions of hemoglobin subunit-beta, lactotransferrin, and serotransferrin in HL yak milk increased significantly compared to TL yak milk. Since these components are blood-derived proteins, it suggests that the permeability of mammary gland tight junctions increase in HL yaks, which is consistent with the characteristics of cows in the degeneration stage. In addition, during mammary gland degeneration, the somatic cell count and the apoptosis rate of mammary epithelial cells increase [25]. This study found that HL yaks contained significantly higher levels of several types of keratins in milk than TL yaks (Appendix A), which may be a result of increased shedding of mammary epithelial cells in HL yak milk since keratin is a marker of epithelial cells [26]. Moreover, cysteine-rich secretory protein 3 (CRISP-3) precursor, which is a key protein in cell apoptosis [27], and vinculin, which can maintain cell growth and differentiation and promote cell survival [28], were also up-regulated and down-regulated in HL yak milk, respectively. This suggests that the mammary glands in HL yaks are at a state of degeneration, consistent with the characteristics of mammary glands of degenerated cows.

During the dry period, the ability of mammary epithelial cells to synthesize lactose, milk fat, casein, α-lactalbumin, and β-lactoglobulin decreases, while the concentration of lactoferrin in mammary secretions increases significantly [4,7]. GO enrichment analysis of DEPs showed that the vesicle and cytoplasmic vesicle were the most abundant categories in the cellular component (Figure 4A). In contrast, the transporter activity was the most abundant category in the molecular function (Figure 4B). Vesicles play a key role in protein transport [29], and the difference in protein transport activity may be related to milk protein components in HL yak milk. The KEGG pathway enrichment analysis, which was based on the DEPs, revealed that the metabolic pathways with the most differences were those that involved carbohydrate metabolism, the biosynthesis of amino acids, and fat digestion and absorption. Previous research found that a high number of proteins in human and ruminant milk serum were related to metabolic processes [30]. In another study, metabolism-related pathways (such as glycolysis/gluconeogenesis and biosynthesis of amino acids) also enriched many differentially expressed whey proteins in human and bovine colostrum and mature milk [31]. The differences in these metabolic pathways were related to the ability of mammary epithelial cells to synthesize lactose, milk fat, and milk protein in HL yak.

In the sample, the pooled skim milk samples of TL yaks and HL yaks were prepared by mixing equal volumes of 15 skim milk samples of corresponding yaks, respectively. This procedure clearly blurs the individual differences between the tested yak females. However, the lactation period between yak groups was similar. The same sample processing method was used in previous studies [32].

## 5. Conclusions

The iTRAQ technique was used to compare skim milk proteins in yaks from standard and naturally extended lactation (TL yaks and HL yaks, respectively). A total of 202 differentially expressed proteins were identified, among which 109 proteins were up-regulated and 93 were down-regulated in HL yaks compared to TL yaks. The GO function annotation and pathway enrichment analysis suggest that HL yaks produce milk with characteristics that reflect the degeneration stage, similar to that of dairy cows.

## Figures and Tables

**Figure 1 animals-12-00391-f001:**
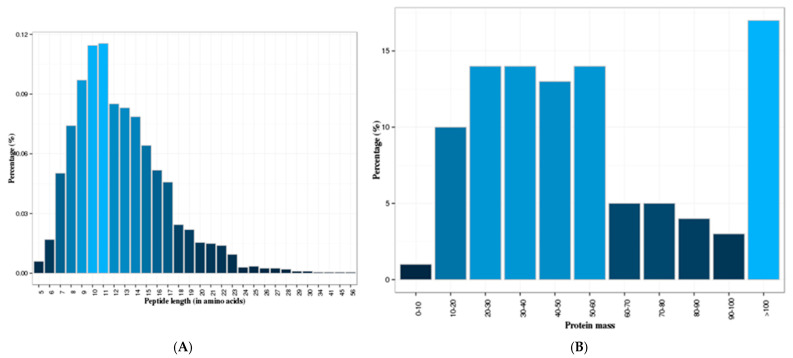
Information on yak skim milk protein identification. (**A**) The proportion of different length peptides. (**B**) The mass distribution of all identified proteins. The *x*-axis indicates the molecular weight (kDa).

**Figure 2 animals-12-00391-f002:**
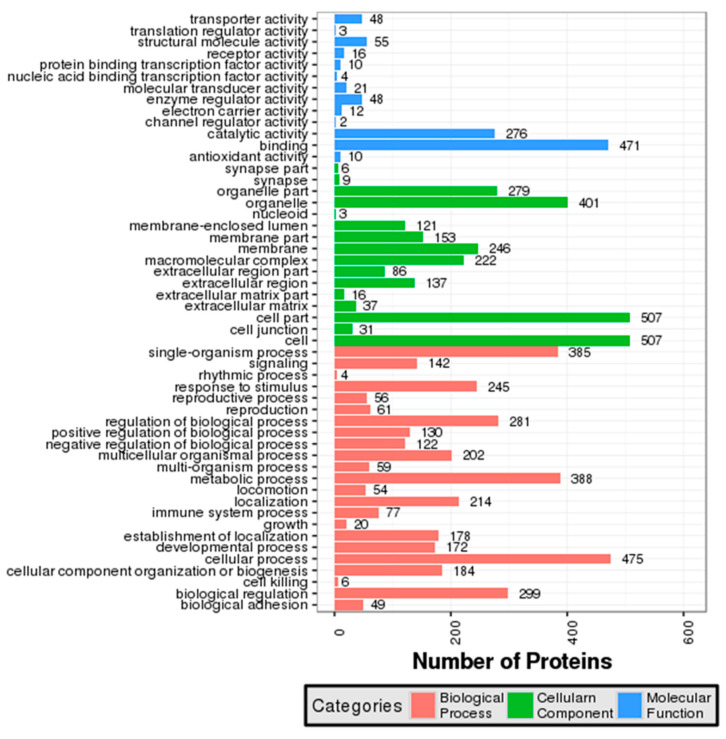
The Gene Ontology analysis of all identified proteins. The bar chart shows the distribution of corresponding GO terms. Different colors stand for different GO terms, and the length represents the GO term’s number of all proteins.

**Figure 3 animals-12-00391-f003:**
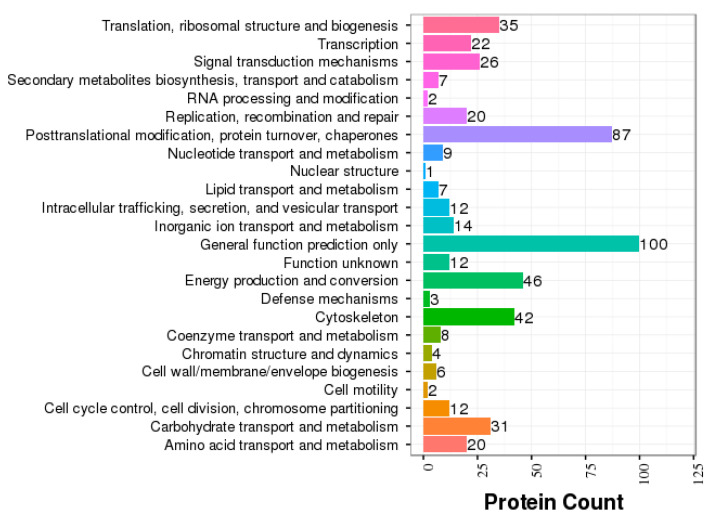
A bar chart of the COG analysis. The *y*-axis displays the COG term, and the *x*-axis displays the corresponding protein count, which illustrates the protein number of different functions.

**Figure 4 animals-12-00391-f004:**
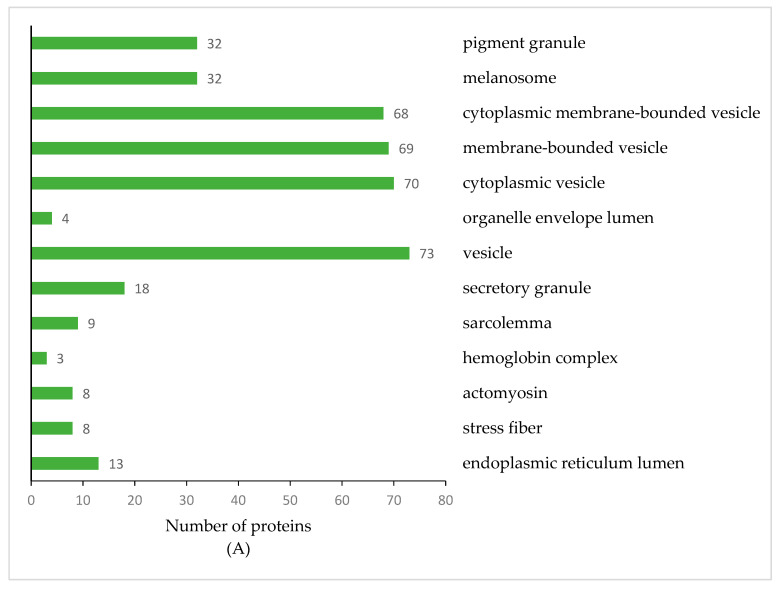
GO Enrichment Analysis of DEPs. Cellular component ((**A**); *p* < 0.05), molecular function classification ((**B**); *p* < 0.05) and biological processes ((**C**); *p* < 0.01) of differentially expressed proteins according to the GO enrichment analysis. The orders of the columns are arranged according to *p*-values (low to high).

**Figure 5 animals-12-00391-f005:**
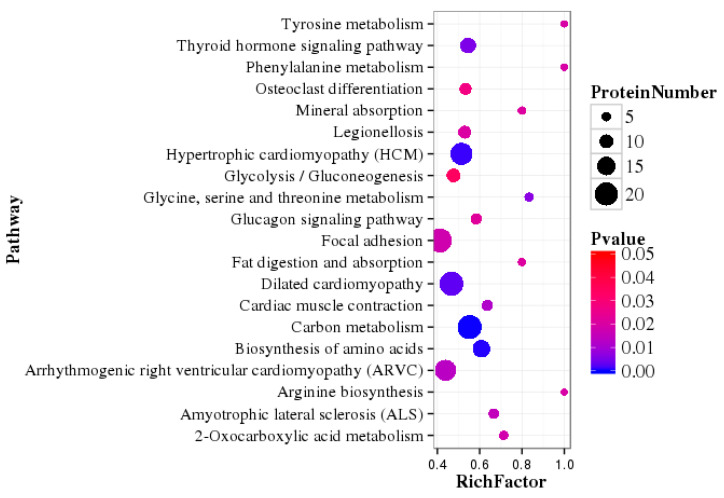
Top 20 pathway enrichment of differentially expressed proteins. Rich Factor is the ratio of the DEPs number annotated in this pathway term to all protein numbers annotated in this pathway term. A greater Rich Factor means greater intensiveness. A lower P value means greater intensiveness.

**Table 1 animals-12-00391-t001:** The top 50 up-regulated skim milk proteins in HL yaks.

No.	NCBInr Accession	NCBInr Description	Mass (kDa)	Protein Coverage (%)	Fold Change *
1	gi|440897559	14-3-3 protein theta, partial	29.1	11.7	10
2	gi|440898761	Spliceosome RNA helicase BAT1	50.5	5.5	6.82
3	gi|440897815	Dolichyl-diphosphooligosaccharide-protein glycosyltransferase 48 kDa subunit, partial	50.8	2.4	6.49
4	gi|440892840	Small nuclear ribonucleoprotein-associated protein N, partial	24.7	3.3	5.73
5	gi|440892885	Protein S100-A1, partial	13.9	11.9	5.56
6	gi|440896162	Tubulin beta-7 chain	50.1	28.6	5.35
7	gi|15718687	40S ribosomal protein S3 isoform 1	26.8	3.7	5.13
8	gi|115498012	Glycogen phosphorylase, liver form	88.3	3.4	4.87
9	gi|440895215	Hemoglobin subunit beta, partial	18.0	38.3	4.64
10	gi|440906467	D-3-phosphoglycerate dehydrogenase	56.8	8.1	4.6
11	gi|440905390	Haptoglobin, partial	45.5	4.3	4.21
12	gi|440907809	Annexin A4, partial	36.1	25.4	4.21
13	gi|32189334	ADP/ATP translocase 2	33.1	10.1	3.90
14	gi|326933110	PREDICTED: histone-binding protein RBBP4-like	47.3	1.9	3.88
15	gi|440895216	Hemoglobin subunit epsilon-1, partial	16.5	23.8	3.87
16	gi|7106387	Proteasome subunit alpha type-5	25.4	5.2	3.79
17	gi|440912614	Annexin A1	39.2	3.8	3.69
18	gi|440903260	Keratin, type I cytoskeletal 42	51.5	5.6	3.58
19	gi|440901472	Secretoglobin family 1D member 2	11.5	8.8	3.45
20	gi|440900805	Sodium/potassium-transporting ATPase subunit alpha-1, partial	115.9	2.6	3.42
21	gi|440907817	Hypothetical protein M91_09610	27.6	4.9	3.4
22	gi|440903642	T-complex protein 1 subunit gamma, partial	64.1	1.9	3.28
23	gi|62460420	Heterogeneousnuclear ribonucleoprotein F	46.0	4.1	3.1
24	gi|440899875	Signal transducer and activator of transcription 1, partial	89.1	1.4	3.06
25	gi|351702463	Calponin-3	19.7	6.2	3.03
26	gi|440902212	Adenosylhomocysteinase, partial	47.5	1.9	2.97
27	gi|444513904	Heterogeneous nuclear ribonucleoprotein A1, partial	20.6	23.9	2.95
28	gi|440902100	Heterogeneous nuclear ribonucleoprotein C	33.8	7.5	2.79
29	gi|73955372	PREDICTED: profilin-1-like	10.5	40.6	2.77
30	gi|440908857	Clathrin heavy chain 1, partial	191.6	5.5	2.76
31	gi|195157178	GL12416	50.8	42.6	2.75
32	gi|440900693	Keratin, type II cytoskeletal 8, partial	54.4	14.8	2.75
33	gi|440901320	Synaptophysin-like protein 1	28.3	4.3	2.69
34	gi|4504447	Heterogeneousnuclear ribonucleoproteins A2/B1 isoform A2	35.8	24.2	2.66
35	gi|115495959	ATP-dependent RNA helicase DDX1	83.1	1.5	2.62
36	gi|440900692	Keratin, type I cytoskeletal 18, partial	49.2	5.0	2.5
37	gi|440900744	Alpha-S2-casein, partial	25.7	21.5	2.48
38	gi|440909809	Hypothetical protein M91_11302, partial	50.1	20.7	2.48
39	gi|344290140	PREDICTED: eukaryotic translation initiation factor 5A-1-like	17.3	7.6	2.45
40	gi|331284195	Nucleolin	77.2	2.0	2.45
41	gi|431895946	NMDA receptor-regulated protein 2	39.3	30.9	2.44
42	gi|114051505	Serpin H1 precursor	46.6	19.9	2.44
43	gi|440899031	Heat shock protein HSP 90-alpha, partial	37.7	29.1	2.36
44	gi|4505585	Platelet-activating factor acetylhydrolase IB subunit beta isoform a	25.7	3.9	2.36
45	gi|440900746	Alpha-S1-casein, partial	24.5	91.6	2.35
46	gi|440908101	L-lactate dehydrogenase B chain	37.4	13.6	2.33
47	gi|440895669	Triosephosphate isomerase	30.9	12.9	2.31
48	gi|440900959	60S ribosomal protein L12, partial	21.2	7.8	2.26
49	gi|78369302	Catalase	60.1	3.0	2.21
50	gi|440904790	Annexin A6, partial	75.7	7.9	2.18

* Relative abundance of proteins in HL yak milk versus TL yak milk.

**Table 2 animals-12-00391-t002:** The top 50 down-regulated skim milk proteins of HL yaks.

No.	NCBInr Accession	NCBInr Description	Mass (kDa)	Protein Coverage (%)	Fold Change *
1	gi|440897611	6-phosphofructokinase, muscle type	94.3	2.8	0.1
2	gi|311257294	PREDICTED: cytochrome b-c1 complex subunit Rieske, mitochondrial-like	29.8	2.6	0.1
3	gi|440913466	Hemoglobin subunit alpha-1, partial	16.4	6.0	0.12
4	gi|440904354	Cytochrome c1, heme protein, mitochondrial, partial	31.5	4.9	0.14
5	gi|440893900	Primary amine oxidase, liver isozyme, partial	86.9	1.3	0.17
6	gi|410964723	PREDICTED: integrin alpha-7 isoform 2	129.1	1.0	0.21
7	gi|110350683	Biglycan precursor	41.9	10.8	0.21
8	gi|440901307	Tropomyosin alpha-1 chain	37.7	6.4	0.25
9	gi|440904898	Transgelin, partial	23.0	35.8	0.27
10	gi|114052094	Calponin-1	33.4	36.7	0.31
11	gi|440910551	Platelet glycoprotein 4	53.4	4.2	0.31
12	gi|440897546	Phosphate carrier protein, mitochondrial, partial	42.3	5.1	0.32
13	gi|109891934	RecName: Full=Isocitrate dehydrogenase	51.1	6.9	0.34
14	gi|440893758	PDZ and LIM domain protein 3	39.9	4.1	0.35
15	gi|440910545	Tropomyosin alpha-1 chain	37.4	9.2	0.36
16	gi|115497074	S-formylglutathione hydrolase	32.3	2.5	0.36
17	gi|440903063	2-oxoglutaratedehydrogenase, mitochondrial	118.6	1.5	0.39
18	gi|440907850	Desmoplakin, partial	329.7	0.5	0.42
19	gi|440907068	Sarcoplasmic/endoplasmic reticulum calcium ATPase 1, partial	111.4	2.1	0.42
20	gi|440911054	Desmin, partial	46.3	31.2	0.43
21	gi|78369436	3-ketoacyl-CoA thiolase, mitochondrial	41.9	5.4	0.45
22	gi|440909063	Aspartate aminotransferase, cytoplasmic, partial	47.2	2.4	0.47
23	gi|431894354	Myosin regulatory light polypeptide 9	20.2	39.2	0.47
24	gi|426220158	PREDICTED: tropomodulin-1	40.5	2.2	0.47
25	gi|332634684	Glutamate [NMDA] receptor subunit epsilon-3 precursor	94.2	1.7	0.51
26	gi|440896699	Four and a half LIM domains protein 2, partial	34.3	3.9	0.51
27	gi|344253851	Actin, gamma-enteric smooth muscle	42.2	59.0	0.51
28	gi|440904982	Sodium-dependent phosphate transport protein 2B	77.3	2.3	0.52
29	gi|440897772	Filamin-C	293.4	2.1	0.52
30	gi|440904421	ATP-binding cassette sub-family G member 2	73.6	1.5	0.52
31	gi|440897862	Filamin-A	283.1	18.1	0.53
32	gi|440896802	Myosin-11, partial	176.4	4.3	0.55
33	gi|440899469	Citratesynthase, mitochondrial	52.0	2.4	0.55
34	gi|74354891	GLUD1 protein	57.4	1.8	0.56
35	gi|120419518	enigma protein	50.8	4.8	0.56
36	gi|440898241	NADH-ubiquinone oxidoreductase 75 kDa subunit, mitochondrial, partial	80.5	1.8	0.56
37	gi|440898951	Creatine kinase B-type, partial	43.2	18.2	0.58
38	gi|444518229	Myosin light chain 6B	17.1	23.2	0.58
39	gi|440912794	Calcium-binding mitochondrial carrier protein Aralar1, partial	73.9	1.5	0.59
40	gi|440901372	Fibrinogen alpha chain, partial	86.6	12.8	0.59
41	gi|114053019	alpha-1B-glycoprotein precursor	36.3	12.9	0.59
42	gi|119888979	PREDICTED: basement membrane-specific heparan sulfate proteoglycan core protein	474.8	1.2	0.59
43	gi|440909830	Factor XIIa inhibitor, partial	56.1	15.3	0.61
44	gi|440902649	Aspartate aminotransferase, mitochondrial	47.9	2.3	0.61
45	gi|440893207	Myosin-7, partial	223.1	4.9	0.61
46	gi|77404217	Phosphoglycerate mutase 1	28.9	10.2	0.63
47	gi|296489917	TPA: cellular repressor of E1A-stimulated genes	17.5	22.9	0.64
48	gi|139948177	Cysteine and glycine-rich protein 1	21.4	25.9	0.64
49	gi|410932519	PREDICTED: uncharacterized protein LOC101077311	11.4	35.9	0.65
50	gi|440901371	Fibrinogen beta chain, partial	56.7	19.8	0.65

* Relative abundance of proteins in HL yak milk versus TL yak milk.

## Data Availability

Not applicable.

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
