# Peer review of "Comparative Proteomics Study of Yak Milk from Standard and Naturally Extended Lactation Using iTRAQ Technique"

_animals, 2022, doi:10.3390/ani12030391_

Round 1
Reviewer 1 Report
The paper entitled Comparative proteomics study of yak milks from standard and naturally extended lactation using iTRAQ technique is really interesting because it adds new insights in the field of animal milk proteomics and in particular to YAK milk proteomics.Yak is very important of some area of Tibe- China, infact it represents one of the main unique livestock species in the ethnic minority areas of this zone. Moreover, it provides important living materials for local farmers. There are some minor comments to this paper.
First: do not use the plural of milk. Remove s for word milk.
Second. There is an important review that summarizes the technique and method employed in animal milk proteomics. Please cite it and add a few words about the technique that you used with respect to this review. 10.1016/j.jprot.2012.05.028
Third. Did the authors investigate eventually of potentially not-allergenic properties of YAK milk? If not, it is important in the introduction to refer to this important field in the study of milk proteomics. Please read and cite two important papers regarding proteins allergies. 10.1007/s12016-020-08810-9 and 10.1002/mnfr.201700278
- The research is important because is able to investigate proteomics of YAK milk.
- The topic is really original, because YAK is a minor species but important for economy of the China- Tibet.
- The paper is well written and clear
- They address the comparison of YAK milk from naturally and extended lactation period through I TRAQ proteomics, that elucidate also new insights of physiology of animals.
Author Response
Dear reviewer:
We would like to thank you for your careful reading, helpful comments, and constructive suggestions, which has significantly improved the presentation of our manuscript.
We have carefully considered all comments from the reviewers and revised our manuscript accordingly. The manuscript has also been double-checked, and the typos and grammar errors we found have been corrected. In the following section, we summarize our responses to each comment from the reviewers. We believe that our responses have well addressed all concerns from the reviewers. We hope our revised manuscript can be accepted for publication.
In our revisions, we paid specific attention to 1) Remove s for word milk in the full.
2) This paper is referred to in the introduction and methods. (doi: 10.1016/j.jprot.2012.05.028)
3) About potentially not-allergenic properties of YAK milk, the main allergens in milk is β-lactoglobulin and was no significant difference in the milk of HL yaks compared with those of TL yaks. Therefore, we did not consider this part.
Happy New Year!
Reviewer 2 Report
The paper of Zheng et marks an important goal in the field of milk proteomics of minor species (YAK). The workflow is appropriate as well as methodology reported, together with the conclusion. My minor comment is devoted to the necessity to improve references, in particular in the 2.3 paragraph please refer to doi 10.1016/j.ijfoodmicro.2019.108265 in KEGG analysis.
Author Response
Dear reviewer:
We would like to thank you for your careful reading, helpful comments, and constructive suggestions, which has significantly improved the presentation of our manuscript.
We have carefully considered all comments from the reviewers and revised our manuscript accordingly. The manuscript has also been double-checked, and the typos and grammar errors we found have been corrected. In the following section, we summarize our responses to each comment from the reviewers. We believe that our responses have well addressed all concerns from the reviewers. We hope our revised manuscript can be accepted for publication.
In our revisions, we paid specific attention to 1) This paper is referred to in the introduction and methods. (doi: 10.1016/j.jprot.2012.05.028)
2) This paper is referred to the 2.3 paragraph. (doi 10.1016/j.ijfoodmicro.2019.108265).
3) In the sample, the pooled skim milk samples of TL yaks and HL yaks were prepared by mixing equal volume of 15 skim milk samples of corresponding yaks, respectively. This procedure clearly blurs the individual differences between the tested yak females, But the lactation period of yaks between groups was similar. The same sample processing method was used in previous studies .( doi: 10.3389/fnut.2021.670099 )
Happy New Year!
Reviewer 3 Report
Dear Authors,
I have found your work very interesting. The extended lactation is a common phenomenon not only in yaks but also n other herbivores, mainly natural conditions. Milk protein compositions during this period could be of interest not only of scientists but also of farmers and breeders of yaks. Using the iTRAQ method was the good idea to compare different types of proteins. I think that additional materials are extremely valuable source of knowledge about the number and functions of studied proteins.
I am only worried about the mixing of mixing milk of 15 yaks to obtain 1 sample of milk. This procedure clearly blurs the individual differences between the tested yak females. I understand that the analysis of all proteins in all tested animals would be very complicated and extremely difficult to perform, but it seems to me that mixing the milk samples excludes the influence of a single individual. It is only the concern.
I think or even suggest that conclusion should be enriched in your personal opinion about using the iTRAQ method for your experiment.
Best wisher Reviewer
Author Response
Dear reviewer:
We would like to thank you for your careful reading, helpful comments, and constructive suggestions, which has significantly improved the presentation of our manuscript.
We have carefully considered all comments from the reviewers and revised our manuscript accordingly. The manuscript has also been double-checked, and the typos and grammar errors we found have been corrected. In the following section, we summarize our responses to each comment from the reviewers. We believe that our responses have well addressed all concerns from the reviewers. We hope our revised manuscript can be accepted for publication.
In the sample, the pooled skim milk samples of TL yaks and HL yaks were prepared by mixing equal volume of 15 skim milk samples of corresponding yaks, respectively. This procedure clearly blurs the individual differences between the tested yak females, But the lactation period of yaks between groups was similar. The same sample processing method was used in previous studies .( doi: 10.3389/fnut.2021.670099 ). We inserted the idea into the discussion.
Happy New Year!